# Applications of Circulating Tumor Cells and Circulating Tumor DNA in Precision Oncology for Breast Cancers

**DOI:** 10.3390/ijms23147843

**Published:** 2022-07-16

**Authors:** Sridevi Addanki, Salyna Meas, Vanessa Nicole Sarli, Balraj Singh, Anthony Lucci

**Affiliations:** 1Department of Surgical Oncology, The University of Texas MD Anderson Cancer Center, Houston, TX 77030, USA; saddanki@mdanderson.org; 2Morgan Welch Inflammatory Breast Cancer Research Program and Clinic, The University of Texas MD Anderson Cancer Center, Houston, TX 77030, USA; smeas@mdanderson.org (S.M.); vnsarli@mdanderson.org (V.N.S.); bsingh@mdanderson.org (B.S.); 3Department of Breast Surgical Oncology, The University of Texas MD Anderson Cancer Center, Houston, TX 77030, USA

**Keywords:** liquid biopsy, early stage breast cancer, late-stage breast cancer, circulating tumor cells, circulating tumor DNA, next generation sequencing, diagnosis, screening

## Abstract

Liquid biopsies allow for the detection of cancer biomarkers such as circulating tumor cells (CTCs) and circulating tumor DNA (ctDNA). Elevated levels of these biomarkers during cancer treatment could potentially serve as indicators of cancer progression and shed light on the mechanisms of metastasis and therapy resistance. Thus, liquid biopsies serve as tools for cancer detection and monitoring through a simple, non-invasive blood draw, allowing multiple longitudinal sampling. These circulating markers have significant prospects for use in assessing patients’ prognosis, monitoring response to therapy, and developing precision medicine. In addition, single-cell omics of these liquid biopsy markers can be potential tools for identifying tumor heterogeneity and plasticity as well as novel therapeutic targets. In this review, we focus on our current understanding of circulating tumor biomarkers, especially in breast cancer, and the scope of novel sequencing technologies and diagnostic methods for better prognostication and patient stratification to improve patient outcomes.

## 1. Introduction

Breast cancer is the most prevalent malignancy among women in the United States, with an estimated 287,850 newly diagnosed cases in 2022 [1]. It is projected that around 43,250 women will die of the disease this year, making it the second biggest cause of cancer-related fatalities in women. Breast cancer is a complex disease, in which incidence increases with age as a result of the accumulation of somatic mutations in the mammary glands. Malignancy in the breast tissue is heterogeneous and broadly divided into different subtypes based on the molecular aberrations present in the tumor [2].

In the luminal A subtype, which accounts for 50–60% of breast cancer cases, the tumors are positive for estrogen receptor (ER) and/or progesterone receptor (PR) [3]. Because of lower levels of Ki67, a protein associated with the growth of the cancer cells, the luminal A subtype is low grade, less aggressive, and carries a good prognosis. Another subtype of breast cancer is characterized by the presence and overexpression of human epidermal growth factor receptor 2 (HER2). HER2-enriched tumors tend to grow faster and could have a worse prognosis if not for the effectiveness of HER2-targeted therapies such as trastuzumab and pertuzumab. The luminal B breast cancer subtype is also hormone receptor-positive, with the presence of ER and/or PR, but has higher levels of Ki67 and may also be HER2-positive [4]. Luminal B patients’ prognosis is slightly worse than that of luminal A patients.

The most ominous subtype is triple-negative breast cancer (TNBC), characterized by the lack of ER and PR expression and HER2 amplification [5]. TNBCs account for 15–20% of invasive breast cancers and are predominantly high-grade, high-risk tumors that occur more frequently in young women, especially Black women [6]. Inter- and intratumoral heterogeneity contribute to the aggressive nature of this subtype. Women presenting with TNBC have a higher rate of distant recurrence and poorer prognosis than do women with other subtypes of breast cancer [7].

Breast cancer is thus a heterogeneous and complex disease with a wide range of histologic characteristics, treatment responses, metastatic activity, and patient outcomes. Apart from age and genetic, hormonal, and reproductive components of breast cancer risk, there are modifiable factors such as excess body weight, physical inactivity, alcohol use, and receipt of hormone replacement therapy. Breast cancer incidence in women in the United States increased by an average of 0.3% per year between 2004 and 2018 [8]. Incidentally, the incidence of breast cancer increased among women 20–49 years old while it decreased in women older than 50 years. The US Preventive Services Task Force recommends mammography screening every 2 years for women 50–74 years old. However, a screening study including 993,000 people found that screening did not affect the incidence of stage IV illness [9]. In addition, although early stage cancer detection facilitates curative surgical resection in many solid tumors, including breast cancers, mammography is limited by its poor sensitivity, overdiagnosis, false-positive rates, and the discomfort and anxiety it causes patients. Thus, a better prognosis of breast cancer is impeded by a lack of early screening programs in young women and effective diagnostic tools in general [10,11]. These limitations highlight the importance of developing new technologies and strategies for the early identification and treatment of breast cancer.

Liquid biopsies are emerging as a minimally invasive method for early detection and risk management of breast cancer. The purpose of this review is to examine the utility of liquid biopsy indicators of breast cancer, circulating tumor cells (CTCs), and circulating tumor DNA (ctDNA), as well as current improvements and technological advancements in the field.

## 2. Current Understanding and Utility of Liquid Biopsy Markers

Tumor biopsies have been the standard method for tumor tissue detection thus far. Tissue biopsy specimens are collected from primary or metastatic tumor sites to generate a histopathological and genetic profile, determine the patient’s prognosis, and guide therapy selection. However, traditional tissue biopsies are limited by their invasive nature, potential risk linked with tumor site, expense of the procedure, difficulties in accessing certain sites, and processing time. In addition, they are insufficient to portray the complete genomic picture and heterogeneity of the tumor. These limitations necessitate the development of alternative blood-based markers for therapeutic use.

Liquid biopsies were developed as a minimally invasive and less expensive alternative to tissue biopsies [12,13]. Liquid biopsies sample tumor-originated material obtained from the peripheral blood or other bodily fluids and examine circulating biomarkers such as CTCs, cell-free DNA (cfDNA), microRNA, cell-free RNA (cfRNA), exosomes, extracellular vesicles, methylated genes, proteins, metabolites, and tumor-educated platelets. Even though tumor-derived molecules are found in bodily fluids such as urine, cerebrospinal fluid, ascites, and multiple effusions, blood is the principal resource for effective separation and subsequent molecular analysis of circulating tumor material. Unlike tissue biopsies, liquid biopsies can capture the tumor’s heterogeneity while allowing rapid sample processing at a much lower cost. With the advent of new technologies, there has been tremendous progress in using liquid biopsy markers for early detection and screening, prognosis, early detection of relapse, longitudinal sampling, and real-time monitoring of therapy effectiveness. In this review, we focus on two particular circulating biomarkers, ctDNA and CTCS, in early and late-stage breast cancer and analyze their potential clinical value.

## 3. Circulating Tumor Cells and Circulating Tumor DNA

The first blood-based circulating tumor marker to be discovered was CTCs in 1869, found in patients with metastatic cancer; circulating nucleic acids were discovered in 1948 [14,15]. Later, cfDNA was uncovered and demonstrated to identify specific mutations [16,17,18]. During the growth of a primary tumor and later its metastatic dissemination, a number of its components are shed into circulation following events such as apoptosis, necrosis, etc. [19]. These components are primarily composed of CTCs, ctDNA, cfRNA, exosomes, tumor-educated platelets, and extracellular vesicles, broadly termed the tumor circulome [13]. Of these circulating tumor markers, CTCs and cfDNA have gained prominence in the pursuit of developing diagnostic tools for clinical application. Various studies have described the value of CTCs and ctDNA for predicting a patient’s prognosis and response to therapy.

CTCs are disseminated cancer cells from the primary or metastatic tumor(s) that are found in the peripheral blood. Their origin from multiple tumor sites, some of which may undergo the phenomenon of epithelial-to-mesenchymal transition (EMT), makes CTCs a diverse population reflecting the spatiotemporal heterogeneity of the disease [20]. CTC identification necessitates particular enrichment methods based on physical or biological features. For example, the size, density, or deformability of the cells and the presence of an electric charge are used in physical approaches. Protein secretion and cell surface antigen expression are used for biological methods. The US FDA licensed the most frequently used CTC enumeration platform, CellSearch (Menarini-Silicon Biosystems), for clinical use in breast cancer patients in 2004.

ctDNA is fragmented DNA originating from the tumor and released into the blood; it is a subset of cfDNA, which can originate from any cell in the body. ctDNA, like other circulating blood indicators, can be produced by apoptosis, necrosis, or active shedding of tumor cells. However, apoptosis is the most common method, and hence cleaved DNA is usually around 140–180 bp in length [19,21,22]. The amount of ctDNA found in bodily fluids depends on the tumor burden and tumor proliferation and can reveal genomic aberrations, including copy number variants, alterations of methylation patterns, point mutations, microsatellite alterations, chromosomal rearrangements, etc. [23]. Circulating DNA is quickly cleared from the blood and has a half-life of 15 min to 2 h, thus making it a dynamic biomarker for monitoring tumor burden [24]. The most extensively used methods for detecting ctDNA are PCR-based approaches and next-generation sequencing (NGS)-based technologies [25,26]. The NGS workflow principally includes DNA ligation, library construction, clonal amplification of the template, and massive parallel sequencing of the millions of DNA fragments derived from the samples [26].

PCR-based ctDNA detection requires knowledge of the mutations or alterations to be measured, mostly by tissue biopsies or frequently targeted hotspot mutations. Whole-genome and whole-exome sequencing are used in untargeted techniques. Some PCR techniques include digital drop polymerase chain reaction (ddPCR), amplification refractory mutation system PCR, peptide nucleic acid/locked nucleic acid-mediated PCR, pyro phosphorolysis-activated polymerization, and beads, emulsion, amplification, and magnetics (BEAMing) [25,27,28,29,30].

## 4. CTCs in Early Stage Breast Cancer

Early stage breast cancer is more commonly diagnosed than more advanced breast cancer, but around 20% of these patients have recurrence [31]. CTCs in the early stages of breast cancer can generate micrometastases and thus are the seeds of the metastatic cascade. CTCs may also serve as surrogate markers for minimal residual disease (MRD).

Many researchers have looked at the detection and characterization of CTCs in early stage breast cancer peripheral blood. For HER2-positive patients with primary breast cancer, the GeparQuattro clinical study examined neoadjuvant chemotherapy (NACT) that included trastuzumab [32]. The researchers concluded that CTC numbers are low in early stage disease using the FDA-approved CellSearch system for CTC detection. Despite a drop in CTC levels following NACT, there was no correlation between persisting CTC levels and treatment response. They designed HER2 immunoscoring of CTCs to direct patients whose CTCs overexpress HER2 to receive HER2-targeted therapies.

Another group evaluated the prognostic significance of cytokeratin (CK-19)-positive CTCs in early stage breast cancer after NACT and identified it to be an independent risk factor [33]. Sandri et al. examined the possible role of CTCs in operable breast cancer and determined that 30% of patients have CTCs before and after surgery [34] Another study looked into the prognostic value of CTCs in early stage breast cancer and found that the persistence of CTCs before and after NACT identifies a patient subpopulation linked with a higher risk of recurrence [35]. Pierga and colleagues determined that CTCs enable the prediction of early metastatic relapse following NACT in large operable and locally advanced breast cancer [36]. The same group examined the clinical outcomes of CTC detection in non-metastatic breast cancer patients and reported that detecting ≥1 CTC/7.5 mL before NACT accurately predicts overall survival (OS) [37]. This study was followed up by performing CTC counts on 118 patients before and after chemotherapy and examining survival. It was concluded that CTC detection is independently associated with significantly worse outcomes, especially 3–4 years after surgery [37]. Our research group looked at CTC data from chemotherapy-naïve patients with stage I-III breast cancer at definitive surgery and discovered that having ≥1 CTC/7.5 mL predicts early recurrence and a shorter OS [38]. We then investigated the presence of CTCs after NACT in stage I-III TNBC and concluded that ≥1 CTC is predictive of relapse and survival [39].

A large prospective trial of primary breast cancer patients revealed the independent prognostic value of CTCs before and after NACT [40]. The BEVERLY-2 trial evaluated the safety and efficacy of NACT with bevacizumab and trastuzumab to treat patients with HER2-positive inflammatory breast cancer (IBC) [41]. In the prospective survival analysis at 3 years of follow-up, CTC analysis predicted 81% vs. 43% percent disease-free survival (DFS) for patients with ≥1 CTC/7.5 mL of blood at baseline. CTC detection was also found to be a strong and independent predictor of survival in patients with nonmetastatic IBC in the BEVERLY-1 and BEVERLY-2 trials [42]. When the group of patients with pathologic complete response to NACT was merged with the group that had no CTC detection at baseline, a subgroup of IBC patients with a 3-year OS of 94% was discovered. As a result of the BEVERLY study, the role of CTCs in tumor spread and their potential application for IBC patient stratification were established.

Interestingly, in a secondary analysis performed on a randomized clinical trial of patients with localized breast cancer for five or more years following diagnosis, detection of CTCs was related to a greater probability of recurrence in HR-positive patients [43]. Following the discovery of CTCs as a predictive marker for death and recurrence in breast cancer, Goodman et al. looked into the function of CTC status in predicting local recurrence or the survival benefit of adjuvant radiotherapy in early stage breast cancer [44]. They discovered that patients who had at least one CTC treated with radiotherapy had significantly higher recurrence-free, disease-free, and overall survival than those who did not have CTCs. As a result, CTC status may be an essential predictor of radiotherapy benefits in patients with early stage breast cancer. The SUCCESS trial provided more information on the prognostic relevance of CTCs to follow-up care in high-risk early stage breast cancer. The occurrence of CTCs 2 years after treatment was linked to a lower OS and DFS [45]. Table 1 summarizes the articles about CTCs in early stage breast cancer.

## 5. CtDNA in Early Stage Breast Cancer

Many approaches have been used in the past decade to detect and measure ctDNA in patients with early stage breast cancer. For example, Beaver et al. detected ctDNA in the plasma of early stage breast cancer patients using primary breast tumors with matched pre- and post-surgery samples [47]. PIK3CA mutations identified using Sanger sequencing in tumor tissue were accurately detected in plasma samples using ddPCR.

Researchers have assessed ctDNA amounts in patients with early stage breast cancer for the purpose of reducing mortality by early detection and therapy modification. Riva et al. investigated the presence of ctDNA in a cohort of patients with nonmetastatic TNBC to examine whether ctDNA was associated with response to NACT and measure MRD after surgery [48]. Using ddPCR, they assessed ctDNA presence at baseline with a detection rate of 75%. Furthermore, there was a rapid decline in ctDNA levels during NACT as well as undetectable MRD. The researchers also found that a slow reduction in ctDNA levels during NACT was substantially linked to a shorter survival time. Phallen and colleagues sought the detection of early stage cancers using ctDNA [49]. They developed targeted error correction sequencing (TEC-seq), which allows direct examination of sequence changes in cfDNA using massive parallel sequencing. Fifty-eight cancer-related genes were investigated by this method, and somatic mutations were discovered in the plasma of 71% of the early stage breast cancer patients. There was a great degree of concordance between mutations found in the tumor samples and ctDNA. CancerSEEK is a blood test aimed at detecting eight different cancer types, including nonmetastatic breast cancer, by assessing the levels of circulating proteins and tumor-specific mutations in the circulating DNA [50]. 

A principal caveat of tissue biopsies is their inability to track changing genomic profiles over time, which liquid biopsies can overcome through serial sampling. Longitudinal fluid biopsy sampling allows precise monitoring of therapeutic efficacy and tracks the development of treatment resistance. Many recent studies have attempted sequencing ctDNA to generate mutation profiles to identify gene alterations in the resistant clones. Such mutation tracking showed ctDNA to be associated with relapse in early stage breast cancer [51,52,53]. ddPCR was used to track mutations discovered in the primary tumor for their presence in ctDNA in post-surgery and follow-up samples. Using massive parallel sequencing, the mutational profile of ctDNA was used to identify the genetic features of therapy-resistant tumor clones. The group also monitored early stage breast cancer with a lead time of ctDNA detection of 10.7 months following disease relapse [53]. These investigations elucidated the relevance of ctDNA in monitoring tumor burden and tracking the emergence of resistant clones to enable appropriate therapy selection.

Rothe and colleagues looked at the relationship between ctDNA and response to anti-HER2 therapy and discovered that HER-2-enriched tumors with no ctDNA had the greatest pathologic complete response rates at baseline, indicating that ctDNA can be used as a biomarker for NACT response in HER-2-amplified breast cancer [54]. Along these lines, Zhang et al. investigated the genomic variants of ctDNA for their potential use as actionable biomarkers in early stage breast cancer treatment [55]. Deep sequencing of plasma and matching tissue samples revealed that the intratumoral heterogeneity found in tumor tissues was reflected in ctDNA values. Furthermore, post-surgery ctDNA positivity was linked to a higher percentage of lymph node metastasis, indicating the possibility of recurrence and distant metastasis.

Researchers also looked into the use of ctDNA analysis for diagnosing early stage breast cancer after mammography results [56]. They analyzed primary breast tissue with the Illumina NGS TruSeq Custom Amplicon Low Input Panel and plasma with SafeSEQ (Sysmex Inostics). Additional ctDNA mutations in the TP53 and PIK3CA genes were discovered in the sequencing data that were not identified in the tissue specimens. Furthermore, age, tumor grade and size, immunohistochemistry subtype, Breast Imaging Reporting and Data System classification (BI-RADS) category, and lymph node positivity were all linked to ctDNA mutations.

Another study examined the efficacy of ctDNA to predict relapse in TNBC patients with residual disease after NACT [57]. Using the Oncomine NGS panel for ctDNA sequencing, the researchers demonstrated that recurrence in such patients may be predicted with high specificity but modest sensitivity. Moreover, recurrence was quick in the event of ctDNA detection. Diagnostic methods to correctly predict residual disease following NACT are needed in localized breast tumors. NACT can help guide treatment decisions such as the extent of surgical resection and the need for radiation treatment. Because present diagnostic techniques lack sensitivity, a therapy monitoring biomarker that can accurately discriminate residual disease from disease elimination would allow patients to obtain tailored therapy [58,59]. McDonald and colleagues created the targeted digital sequencing (TARDIS) method for multiplexed analysis of patient-specific cancer mutations [60]. This approach proved successful in detecting minute amounts of residual DNA in patients’ plasma. The researchers discovered that patients who obtained pathologic complete response had lower ctDNA concentrations than those with residual disease. In addition, during NACT, the drop in ctDNA levels was more pronounced in the group with pathologic complete response. Table 2 summarizes studies relevant to ctDNA in early stage breast cancer.

## 6. CTCs in Metastatic Breast Cancer

Although significant strides have been made toward improving breast cancer survival rates, resistance to treatment develops in many patients and eventually leads to death from metastatic breast cancer. CTCs are released into the bloodstream of patients with solid tumors, functioning as seeds for subsequent metastasis. Elevated levels of CTCs during cancer treatment are an indicator of cancer progression and therefore can reveal the mechanisms of metastasis. CTCs in metastatic breast cancer provide more information than those in early stage breast cancer because they reflect the dominant clones at metastatic homing sites and aid in quantifying the remaining tumor burden. Thus, CTCs are invaluable tools for cancer detection and monitoring through a simple, non-invasive blood draw, allowing multiple longitudinal sampling.

In early efforts to evaluate the predictive values of CTCs in metastatic breast cancer, CTCs were indeed revealed to be a strong independent prognostic marker for the disease [61]. Martin et al. sought to analyze the relationship between OS and CTC counts after the first round of chemotherapy [62]. They acquired CTC counts at the baseline, before starting the first cycle of chemotherapy, and after the first cycle of chemotherapy to examine the prognostic relevance of CTC measures before giving the second round of chemotherapy. The CTCs were separated into low count (0–4 CTCs) and high count (≥5 CTCs). Patients with 0–4 CTCs after the first chemotherapy cycle had a significantly better OS (median OS: 38.5 months vs. 8.7 months), PFS (median 9.4 vs. 3.0 months), and clinical benefit rate (77% vs. 44%) than patients with ≥5 CTCs. In conclusion, the researchers determined that CTC measures following the first chemotherapy cycle were an early and robust predictor of treatment outcomes in metastatic breast cancer patients.

A pooled analysis of individual patient data was acquired from 17 European centers to evaluate the clinical validity of CTC quantification in the metastatic breast cancer prognosis [63]. Using a 1944 eligible patient database derived from 20 different studies, the researchers found that patients with ≥5 CTCs at baseline had decreased PFS (HR 1.92, 95%, CI 1.73–2.14, *p* < 0.0001) and OS (HR 2.78, 95%, CI 2.42–3.19, *p* < 0.0001) compared with <5 CTCs/7.5 mL plasma at baseline. Furthermore, increased CTC counts 3–5 weeks and 6–8 weeks after treatment were associated with shorter PFS and OS. These data shed light on the independent prognostic value of CTCs for PFS and OS of metastatic breast cancer patients.

Because high CTC levels have been linked to poor prognosis, the SWOGS0500 trial was designed to see if switching chemotherapy in metastatic breast cancer patients with persisting CTCs after the first cycle of first-line chemotherapy would improve OS [64]. After 21 days of chemotherapy, patients with continuously increasing CTCs were randomly assigned to either continue receiving the initial therapy or switch to an alternative chemotherapy regimen. The investigators discovered that switching to an alternate cytotoxic therapy early after the first cycle of chemotherapy did not result in longer OS in individuals with persistently elevated CTCs. Patients with increased or persistent CTCs after first-line chemotherapy may benefit from immunological, targeted, or other therapeutic modalities rather than moving to another type of cytotoxic therapy, according to the findings of this study.

Cristofanilli et al. tested the predictive utility of CTCs for stratifying the patients with stage IV metastatic breast cancer [65]. In a retrospective, pooled analysis based on 18 cohorts, 2436 metastatic breast cancer patients were classified as either stage IV aggressive (≥5 CTCs) or stage IV indolent (<5 CTCs) based on molecular subtype, disease location, and prior treatments. The stage IV indolent group was found to have a longer OS than the stage IV aggressive group. These results demonstrated that CTC levels are a valuable technique for staging and stratifying advanced metastatic breast cancer.

The DETECT study program aimed to assess treatment interventions in metastatic breast cancer patients using CTC phenotypes [64]. The trial’s goal was to compare the safety and quality of life measured by the occurrence of adverse events in patients treated with dual HER2-targeted therapy (trastuzumab plus pertuzumab) plus either endocrine therapy or chemotherapy. It was the first study to categorize participants according to the HER2 phenotype of their CTCs. The HER2 status of the primary tumor was the main criterion for grouping the patients into different DETECT trails, and the clearance of CTCs and PFS eventually estimated the clinical efficacy. In the DETECT III and IV trials, HER2-negative subjects were included, and CTCs were a significant prognostic indicator in these patients. It was reported that the presence of ≥1 CTC with strong HER2 immunostaining was associated with shorter OS. Thus, the study elucidated the biological role of HER2 positivity in CTCs [66].

In HR-positive, HER2-negative metastatic breast cancer patients, the STIC study was designed to assess the efficacy of CTC-driven vs. clinician-driven first-line therapeutic choices [67]. In this randomized, open-label, noninferiority phase 3 trial, patients were grouped into two arms: the CTC-driven arm was given chemotherapy if the CTC counts were ≥5 CTCs/7.5 mL or endocrine therapy if the CTCs were <5 CTCs/7.5 mL, and the clinician decided treatment for the control arm. According to the findings of the STIC trial, a high CTC count (5 CTCs/7.5 mL) indicates a significant negative prognostic factor for OS and PFS. The study results revealed that CTC counts could be reliable biomarkers for selecting the first-line therapy for HR-positive, HER2-negative metastatic breast cancer patients.

Another clinical trial, CirCe01, looked into the efficacy of CTC-based monitoring of patients with metastatic breast cancer after they had completed their third line of treatment [68]. Patients with ≥5 CTCs were randomized to a CTC-driven arm or a standard arm; patients in the CTC arm were assessed after each cycle of therapy, and those whose CTC levels predicted tumor development would be given an alternate line of treatment. However, due to accrual and compliance issues, the trial could not demonstrate the clinical usefulness of CTC monitoring. Studies of CTCs in metastatic breast cancer are summarized in Table 3.

## 7. ctDNA in Metastatic Breast Cancer

In metastatic breast cancer, CTCs can give clues about the genomic landscape of the different tumor populations and the tumor burden. Many studies have recently looked into the prognostic significance of ctDNA. Dawson et al. performed a comparative analysis of conventional serum marker CA15-3, CTCs, and ctDNA in 30 women with metastatic breast cancer receiving systemic therapy. ctDNA was detected in 97% of the patients, and their levels had a greater dynamic range and correlation than CA15-3 and CTCs [24]. Bettegowda et al. used the ddPCR method to detect ctDNA in 640 patients with various cancer types, finding that ctDNA levels were >75% in patients with metastatic breast cancer and 50% in patients with localized breast adenocarcinoma [30]. They also discovered ctDNA in patient samples with no CTCs, indicating that these biomarkers are separate entities. A meta-analysis of 10 eligible studies with 1127 breast cancer patients was conducted to determine the relationship between cfDNA and survival outcomes [69]. The meta-analysis found a robust link between cfDNA and OS (HR 2.41, 95% CI 1.83–3.16) and disease- and recurrence-free survival (HR 2.73, 95% CI 2.04–3.67). These results revealed the predictive and prognostic power of cfDNA in breast cancer.

Shaw et al. examined whether the mutation profiles of cfDNA would capture the heterogeneity exhibited in numerous single CTC profiles [70]. In 112 individuals with metastatic breast cancer, CTCs were counted using CellSearch and compared with matching cfDNA, serum CA15-3, and alkaline phosphatase. Multiple single epithelial cell adhesion molecule (EpCAM)-positive CTCs were recovered by DEPArray in five patients with ≥100 CTCs and compared with matched cfDNA and primary tumor tissue using targeted NGS of about 2200 mutations in 50 cancer genes [70]. Mutational heterogeneity in the PIK3CA, TP53, ESR1, and KRAS genes was mirrored between single CTCs and accurately represented in the cfDNA molecular profiles, highlighting the importance of using cfDNA to monitor the metastatic burden and make treatment decisions. The researchers also compared the efficacy of circulating biomarkers, including cfDNA and CTCs, to traditional breast cancer biomarkers, CA15-3 and AP, in predicting metastatic breast cancer prognosis and treatment response [71]. They concluded that cfDNA levels are the best predictor of disease response and PFS; however, a paired test analyzing both cfDNA and CTC counts provides additional prognostic information and allows patients to be stratified further.

Murtaza et al. found that ctDNA can represent the clonal hierarchy of breast cancer, making it a valuable tool for detecting inter- and intra-metastatic heterogeneity [72]. They performed parallel sequencing of sequential tissue biopsies and plasma ctDNA samples in ER-positive/HER2-positive metastatic breast cancer patients. They discovered that most ctDNA mutations were present in all tumor samples, whereas some rare mutations were only found in one metastatic sample. The PALOMA-3 study combined palbociclib, a CDK4/6 inhibitor, with fulvestrant, a selective ER degrader, to treat women with HR-positive, HER2-negative advanced breast cancer [73]. The study shed light on the early dynamics of ctDNA and established its utility as a biomarker for the CDK4/6 inhibition [74]. Darrigues et al. wanted to see if early changes in ctDNA levels are related to the efficacy of the combination drugs used in the PALOMA-3 trial, which established palbociclib and fulvestrant at the standard of care for ER-positive, HER2-negative metastatic breast cancer [75]. Their findings revealed that serial ctDNA studies prior to radiological evaluation can indeed monitor the efficacy of palbociclib and fulvestrant and that early ctDNA variation is a predictive factor for PFS. A pooled ctDNA analysis was performed, which combined results of 1503 patients from the MONALEESA-2, -5, and -7 trials to identify biomarkers for CDK4/6 inhibition in the advanced breast cancer [76]. The MONALEESA trials looked at the efficacy and safety of ribociclib, a CDK4/6 inhibitor, with a choice of endocrine partners as a first- or second-line treatment for patients with HR-positive, HER2-negative advanced breast cancer. The researchers discovered biomarkers for the response, such as FRS2, MDM2, PRKCA, ERBB2, AKT1, and BRCA1/2, and biomarkers for resistance, such as CHD4, BC11B, ATM, and CDKN2A/2B/2C.

Early ctDNA dynamics were found to be a predictor for PFS in advanced breast cancer in the BEECH trial [77]. The BEECH trial investigated the efficacy of combining capivasertib, an AKT inhibitor, with the first-line chemotherapeutic paclitaxel in metastatic breast cancers that were HER2-positive and HER2-negative and harbored PIK3CA mutations. ctDNA dynamics were assessed as a surrogate for PFS and an early predictor of treatment efficacy. The findings revealed that ctDNA dynamics early during treatment could be used as a proxy for PFS. Additionally, dynamic ctDNA analysis can improve early drug development significantly. The utility of ctDNA analysis to direct therapy in advanced breast cancer was explored by an open-label, multicohort, phase 2a, platform trial of ctDNA testing in 18 UK hospitals [78]. For ctDNA analysis, digital PCR and targeted sequencing were used, with a concordance of 96–99% (*n* = 800, kappa 0.89–0.93). Their findings show that targeted therapies against uncommon HER2 and AKT1 mutations have clinically relevant activity, indicating that these mutations could be used to treat breast cancer. They concluded that with adequate clinical validity for introduction into standard clinical practice, ctDNA testing could provide accurate, quick genotyping that permits the selection of mutation-directed therapy for patients with breast cancer. The study’s results highlight the importance of ctDNA analysis in the development of mutation-directed medicines. Table 4 summarizes articles related to ctDNA in metastatic breast cancer.

## 8. Use of Circulating Tumor Markers for Precision Medicine

Liquid biopsies make longitudinal sampling possible through a patient’s treatment period, provide information on the changing mutation profile of the disease in real time, and serve as predictive markers for precision medicine. NGS-based approaches can be used to build cancer mutation profiles, which can then be used to create patient-specific panels for customized treatment. Coombes and colleagues developed a tailored ctDNA profiling method for detecting breast cancer recurrence [79]. A patient-specific assay was made by employing whole-exome sequencing data to select 16 variations from the primary tumor, which were then evaluated against longitudinal plasma samples to detect ctDNA using ultradeep sequencing. They showed that a patient-specific ctDNA assay could be a specific and sensitive tool for disease surveillance in patients with breast cancer. ESR1 mutations have recently been discovered in the plasma of ER-positive metastatic breast cancer patients, and ESR1-mutated ctDNA has been identified as a predictive marker for response to aromatase inhibitor therapy [80,81]. In addition, mutations in the TP53/PIK3CA genes in ctDNA have been sensitive and specific circulating blood biomarkers, with increasing ctDNA copies related to therapeutic response [24].

Butler et al. conducted whole-exome sequencing on the cfDNA and primary tumor of two metastatic patients [82]. They discovered significant heterogeneity between primary and metastatic disease, and the cfDNA mirrored the metastases. They discovered that the PIK3CA p.H1047R activating mutation is present in primary tumors but not in plasma or metastatic sites. ESR1 mutations were found in the plasma and at the metastatic location but not in the primary tumor [82].

The PI3K/AKT/mTOR pathway is frequently dysregulated in cancer and was shown to play a critical role in tumorigenesis and treatment resistance [83]. The loss of PTEN and PIK3CA gene mutations are the most common genomic events seen in human malignancies, including breast cancer [84]. Genomic aberrations in the PI3K pathway are reported to be increased in metastatic TNBC and suggested as a mechanism of chemotherapy resistance [85,86]. Several preclinical studies underpinned that the presence of PIK3CA mutations are predictive markers of sensitivity to PI3K pathway inhibitors. However, patients with documented PI3K aberrations did not show targeted therapy responses. There is a shortage of validated predictive biomarkers for PI3K pathway inhibitors. ctDNA mutation profiling allows tracking the gain or loss of mutations in the PI3K pathway during cancer evolution and aids in patient-tailored therapy decisions.

PARP inhibitors are synthetically lethal to TNBC tumors harboring BRCA1/2 aberrations by impairing the DNA repair mechanisms [87]. Detecting the genomic alterations through longitudinal plasma sampling enables the identification of resistant genes to PARP inhibitors such as olaparib and veliparib. These studies show that ctDNA detection can be used to follow the molecular alterations for developing strategically targeted therapy.

## 9. Drawbacks of Existing Liquid Biopsy Approaches

Although the field of liquid biopsies is an attractive area of investigation because markers are easily accessible in biological fluids, it is in the process of early development and faces overarching challenges. The main challenge is that most clinical studies of liquid biopsies are retrospective and limited by a small patient cohort. In addition, there is a lack of uniform protocols for sample procurement, storage, and handling as well as for isolation, quantification, and analysis tools. Likewise, standardized methods for biomarker sensitivity and specificity assessment are unavailable. These limitations emphasize the importance of large-scale studies to validate liquid biopsies for clinical application and to standardize protocols and methodologies.

Despite substantial evidence from several analyses that high CTC levels can be an independent prognostic factor, data from the SWOG0500 trial demonstrated that switching chemotherapy in patients with persistent CTC levels has no benefit [64]. Another metastatic breast cancer trial looked into the CTC-guided therapeutic intervention for HR-positive, HER2-negative patients; however, PFS was similar between groups receiving standard and CTC-driven therapies [67]. These studies underscore the need to identify optimal treatment approaches for patients with high CTC levels.

One of the challenges to the implementation of CTC-based liquid biopsies is the heterogeneity of CTCs, which is affected by many factors. For example, CTCs released into the bloodstream from the primary tumor often differ from those released from metastases. Some CTCs may undergo EMT during intravasation and mesenchymal-to-epithelial transition (MET) during the extravasation [88]. During the physiological states of EMT and MET, CTC biomarker expression differs. Thus, CTCs are heterogeneous populations involving subtypes such as intact cells, clusters, and apoptotic cells.

The CellSearch is an immuno-based method for detecting and isolating CTCs from blood and is the only method to have received FDA approval [89]. It involves enrichment of CTCs from whole blood by binding the cells to the anti-EpCAM antibody-conjugated iron nanoparticles, followed by magnetic capture. The cells are further stained by 4′,6-diamidino-2-phenylindole (DAPI) to identify nucleated cells and additionally characterized for epithelial structural cytokeratins such as CK8, CK18, and CK19. Anti-CD45 staining allows the differentiation of CTCs from circulating WBCs. Even though the CellSearch is a routinely used method, since its methodology primarily includes the epithelial marker, EpCAM, CTCs might escape EpCAM-based detection during EMT when the marker levels are significantly reduced. It has been reported that CTCs are enriched in mesenchymal markers (N-Cadherin, TWIST, and Vimentin) and have decreased epithelial marker expression (E-Cadherin, EpCAM, and CK8/18/19) [90]. CTCs exhibiting increased epithelial-mesenchymal(E/M) and mesenchymal(M) phenotypic traits have been demonstrated in metastatic breast cancer [91]. Thus, CTCs undergoing EMT-associated molecular changes cannot be captured by immune-based detection methods solely based on antibodies targeting epithelial markers. An efficient way may be to use a cocktail of antibodies, including epithelial and mesenchymal expression markers.

CTCs also vary in size, typically from 8 to 20 microns in diameter, similar to or smaller than white blood cells but larger than the other blood cells [89,92]. In addition, CTCs are more commonly distinguished from other cells based on their positive expression of EpCAM, cytokeratins, and negative expression of CD45, a leukocyte marker [90,93]. The progress toward the development of CTC-based liquid biopsies for clinical use may be impeded by the existence of such different cell types, which current methods of CTC detection are unable to identify. It has been demonstrated that there are more mesenchymal-type CTCs than epithelial-type CTCs, and the EpCAM-based methods are inadequate to capture such cell populations [94]. Additionally, CTC clusters originating from oligoclonal cells of the primary tumor were shown to increase the metastatic potential by 23- to 50-fold compared with single CTCs [95]. However, current CTC capture methods are not designed to distinguish CTC clusters from single CTCs. Nevertheless, progress is being made. Fu et al. are developing a novel method using a chimeric virus probe which would enable CTC detection with high specificity and sensitivity [96]. The chimeric virus is made of human papillomavirus, enabling specificity toward human CTCs and the SV40-based genome, allowing rapid amplification inside CTCs, thus imparting high sensitivity. If all or most CTCs (including the slow-growing ones) could be tracked by this method, it would overcome the drawbacks of the immune-based CTC detection methods, which are limited by the heterogeneous nature of marker gene expression. It could be helpful in real-time monitoring of the effect of cancer treatments and would likely improve the validity and utility of CTC detection for clinical application.

The rarity of CTCs in billions of blood cells makes isolating high yield, high purity CTCs difficult. CTCs in metastatic breast cancer revealed mainly mesenchymal and epithelial-mesenchymal phenotypes, suggesting that EpCAM- and cytokeratin-based enrichment approaches may not capture cells undergoing EMT [97]. A cocktail of antibodies that includes both epithelial and mesenchymal markers can boost CTC capture efficiency. Unfortunately, bulk sequencing methods cannot accurately reflect the phenotypic diversity of CTCs at various phases of the metastatic cascade. Another complicating factor that leads to false positive results is leucocyte contamination, as it is challenging to collect extremely low-frequency CTCs from high white blood cell populations [98,99]. Because leucocytes and CTCs are similar in size, removing non-target cells using label-free physical and biochemical features may result in a low purity [100]. Recently, two-stage microfluidic chips are enabling selective isolation of CTCs while eliminating leucocytes [101,102]. Highly sensitive and precise separation and detection techniques must be developed to increase the existing utility of liquid biopsy markers.

Compared with CTCs, the prognostic value of ctDNA needs to be further investigated, as few driver mutations have been identified in breast cancer and low amounts of ctDNA have been detected in early stage breast cancer. Additionally, pinpointing the tumor source of circulating DNA fragments remains difficult. While cfDNA levels in a healthy individual can be 3–15 ng/mL of plasma, they can vary from 10 to >1000 ng/mL of plasma in a patient with advanced disease [24]. Even though cfDNA is plentiful in the peripheral blood, the low sensitivity and specificity of cfDNA-based assays make it inappropriate for diagnostic use. The ratio of ctDNA to cfDNA depends on tumor progression, tumor burden, and blood clearance mechanisms. More often, ctDNA constitutes less than 1% of the cfDNA, making its detection challenging in early stage breast cancer [30]. Few sensitive analytical approaches for detecting genetic changes at low allelic frequencies are available. Combining ctDNA with additional biomarkers such as CTCs and imaging technology will likely improve the diagnostic capability of ctDNA.

## 10. Leveraging Next-Generation Sequencing Methods for the Application of Liquid Biopsy Markers toward Personalized Medicine of Breast Cancer

ctDNA sequencing is a non-invasive method for detecting cancer mutations that reduce biopsy-related costs and inconveniences in patients with breast cancer. For sensitive detection of known molecular alterations, methods such as ddPCR and BEAMing are more appropriate. However, ctDNA sequencing using NGS enables a greater detection rate of mutations per patient sample and copy number variations. In addition, NGS of ctDNA is convenient for investigating the presence of genetic aberrations, and this approach is an increasingly favorable method in precision oncology, where tissue biopsies are often inadequate for molecular characterization.

In the past decade, immense progress in the field of NGS has expedited the development of liquid biopsy markers for diagnostic assays. Recent studies indicate that CTCs can be used to predict late recurrence and hence guide treatment selection. However, a major challenge in the field has been the conflicting reports involving CTCs and ctDNA analysis in metastatic breast cancer [103]. These discrepancies are because of the low yields of intact CTCs and bulk sequencing strategies, resulting in the missing of cellular heterogeneity. Nevertheless, efforts have been made to improve CTC yields by developing multiple markers for selection and improved sequencing methods [104]. CTCs provide a snapshot of the genomic abnormalities found in metastatic sites and can be queried to assess mutational changes occurring during the metastatic process and the development of drug resistance [105,106]. Single-cell investigations of patient-derived CTCs revealed genomic changes such as single nucleotide variants, copy number variants, microsatellite instability, and inter- and intrachromosomal rearrangements. Furthermore, single-cell transcriptomics aids in the identification of clinically differentiated tumor subgroups and prognostic biomarkers for the therapy response [107]. As a result, the approach may overcome intratumoral heterogeneity issues and produce distinct lineage patterns for the biomarker development [105].

Emerging technologies such as the single-cell sequencing of CTCs can be a potential tool to identify spatiotemporal tumor heterogeneity, plasticity, and novel pharmacological targets for predicting clinical outcomes and treatment response [105,108]. Single-cell sequencing of CTCs can also reveal tumor evolution through the treatment regimen for the early detection of therapy resistance [109]. Digital measurement of intracellular ER signaling in single CTCs has been found to predict residual disease in patients with localized breast cancer treated with NACT [107]. Furthermore, the 17-gene CTC score suggested that endocrine therapy is insufficient to block ER signaling in functional ESR1-mutant populations, resulting in early metastatic breast cancer progression. A study conducted on circulating breast cancer cells demonstrated that the expression of PD-L1 is highly increased in CTCs in HR-positive, HER2-negative breast cancer patients [110]. These findings emphasize the importance of identifying CTC subpopulations that cause metastasis in order to select patients for therapies such as immune checkpoint inhibition. In addition, single-cell sequencing technology can contribute to a better understanding of the immune system’s complexity and offer new cancer treatment targets [111].

## 11. Integrating CTCs and cfDNA Data for Better Prognosis

Prospective studies of biomarker measurement in large patient cohorts and healthy at-risk individuals will help overcome some of the existing drawbacks of using liquid biopsies for diagnosis and prognosis. Moreover, a combined application of blood-based biomarkers such as cfDNA, ctDNA, and CTCs could benefit their clinical application (Figure 1). The combined analysis of these liquid biopsy indicators for use in routine monitoring procedures has enormous promise. Zhang and colleagues analyzed somatic mutations in plasma ctDNA and matched tumor tissues from early stage breast cancer patients and discovered that combining the cohort’s 74.2% ctDNA detection rate with BI-RADS prediction results increased the predictive value to 92% [55]. Silveira and colleagues looked at the clinical usefulness of analyzing CTCs and ctDNA simultaneously in a prospective biomarker study in HER2-negative patients undergoing first-line chemotherapy [112]. CTC and ctDNA data were evaluated before and after 4 weeks of chemotherapy. The detection profiles and predictive values of circulating biomarkers were assessed to explore their overlap and complementary contribution to metastatic breast cancer management. CTCs and ctDNA showed nonoverlapping detection profiles and complementary prognostic values in metastatic breast cancer.

Keup et al., performed an integrative statistical analysis of multiple liquid biopsy analytes (LBAs) in metastatic breast cancer [113]. To examine the clinical relevance of multiple LBAs, they used matched CTC mRNA, CTC genomic DNA, extracellular vesicle mRNA, and cfDNA and conducted integrated statistical analyses. The study demonstrated the additional benefit of using the combination of LBAs, which increased the prevalence of patients with actionable signals. The researchers concluded that a multiparametric liquid biopsy approach deconvolutes the genomic and transcriptomic complexity for use in clinical practice. Additionally, in a secondary analysis of 196 women with early stage TNBC, the presence of ctDNA and CTCs following NACT was related to significantly lower rates of distant disease-free survival, DFS, and OS [114]. These publications shed light on the use of liquid biopsy markers in conjunction with other diagnostic tools for breast cancer detection.

## 12. Conclusions and Future Directions

Circulating markers have a vast potential in prognosis, therapy monitoring, and guiding precision medicine for patients. However, there is an urgent need to develop more sensitive and accurate diagnostic methods in the early stages of breast cancer when minute liquid biopsy markers are detected. Recent breakthroughs in molecular technologies such as NGS and ddPCR have paved the way for developing microfluidic chip-based cell separation techniques with diagnostic potential. Early detection of breast cancer in its early stages would be conceivable with such solid and sensitive approaches, and current research is headed in that direction. A complementary multi-marker analysis for generating personalized therapeutics is needed to improve liquid biopsy markers’ accuracy and diagnostic capabilities. Liquid biopsies have great potential for prognosis, therapeutic monitoring, and integrative data processing methodologies, and developments in sequencing technologies will speed up the era of precision medicine.

In early stage disease, CTC and ctDNA monitoring will be a significant addition to currently used tools aimed at detecting and subtyping the disease for alignment with therapies. Liquid biopsy monitoring will help identify patients who are at high risk of relapse. Another important element of an overall strategy to reduce the risk of relapse will be novel therapies that are safe and effective, and therefore can be offered as early as possible after diagnosis. Considering a high degree of intratumoral heterogeneity, which is also reflected in liquid biopsies, such therapies will likely be phenotype-based rather than driver-based. In this regard, we have proposed a strategy for modeling a phenotype in cell culture that is crucial in poor prognosis minimal residual disease, i.e., an opportunistic switching of cancer cells between quiescence and proliferation [115,116,117]. Since this phenotype is responsible for the failure of current therapies, noncytotoxic therapies affecting this phenotype would help realize the full potential of liquid biopsies. In late-stage/metastatic disease, liquid biopsy monitoring can be aligned to monitor response to currently offered therapies, some of which are driver-based. To improve outcomes for patients with late-stage disease, therapies designed to inhibit cancer cell plasticity as described above could play a role in inhibiting deep intrinsic resistance. However, disease management will continue to involve a combination of therapies, including cytotoxic therapies.

## Figures and Tables

**Figure 1 ijms-23-07843-f001:**
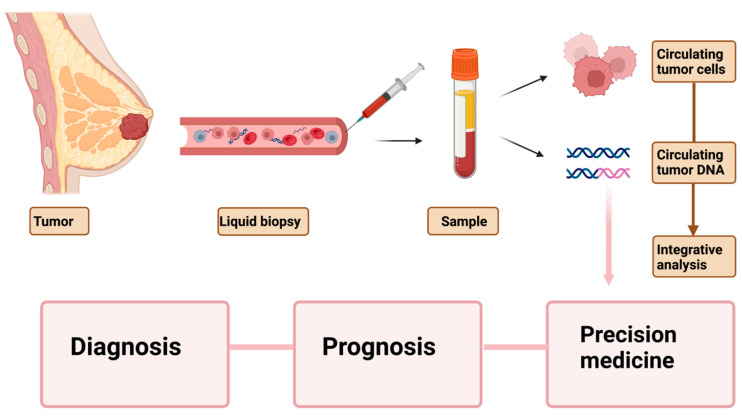
Peripheral blood-based biopsy for breast cancer diagnosis, prognosis, and precision medicine. Circulating blood markers such as CTCs and ctDNA are gaining prominence in cancer diagnosis, therapy monitoring, and development of personalized medicine. However, to bring these liquid biopsy markers into clinical practice, strategies for integrative analyses of these bio-analytes along with existing diagnostic tools need to be developed. Created with BioRender.com.

**Table 1 ijms-23-07843-t001:** CTCs in early breast cancer.

Authors [Reference]	Study Design	Method	Findings
Riethdorf, S., et al., 2010 [32]	CTCs before and after neoadjuvant chemotherapy	CellSearch	1 or more CTC detected in 21.6% patients before NACT. 1 or more CTC detected in 10.6% patients after NACT. No correlation between CTC detection and clinicopathological features. No association between CTC levels and treatment response.
Xenidis, N., et al., 2009 [33]	CK-19 mRNA-positive CTCs before and after adjuvant chemotherapy	RT-PCR	CK-19 mRNA+ CTCs detected in 41% patients before adjuvant chemotherapy. CK-19 mRNA+ CTCs detected in 32.7% patients after adjuvant chemotherapy. CK-19 mRNA+ CTCs associated with more than 3 involved axillary lymph nodes. CK-19 mRNA+ CTCs post-adjuvant chemotherapy associated with chemotherapy-resistant residual disease. CK-19 mRNA+ CTCs post-adjuvant chemotherapy associated with decreased DFS and OS.
Sandri, M.T., et al., 2010 [34]	CTCs before and after surgery	CellSearch	CTCs detected in 30% of patients before and after surgery.CTCs detected pre-surgery associated with positive vascular invasion.
Serrano, M.J., et al., 2012 [35]	CTCs before and after neoadjuvant chemotherapy	Ficoll gradient Immunomagnetic cell separation CK 7, 8, 18, 19, anti-CK FITC	CTCs detected in 71% patients pre-NACT. CTCs detected in 54% patients post-NACT. CTCs detected pre- and post-NACT associated with increased risk of recurrence.
Pierga, J.Y., et al., 2008 [36]	CTCs in patients with large operable or locally advanced breast cancer before and after neoadjuvant chemotherapy	CellSearch	CTCs detected in 23% patients pre-NACT. CTCs detected in 17% patients post-NACT. No association between CTC and primary tumor response. CTCs independently associated with early relapse.
Bidard, F.C., et al., 2013 [37]	CTCs before and after NACT	CellSearch	CTCs detected pre-NACT associated with disease metastatic free survival (DMFS)and OS. CTCs detected post-NACT had no impact. CTCs independently associated with significantly worse outcome during first 3 years of follow-up.
Lucci, A., et al., 2012 [38]	CTCs at time of surgery in chemonaive early stage breast cancer	CellSearch	CTCs detected in 24% of patients at surgery. CTCs associated with decreased PFS and OS.
Hall, C., et al., 2015 [39]	CTCs after NACT in TNBC	CellSearch	CTCs detected in 30% of patients after NACT. CTCs associated with decreased RFS and OS.
Rack, B., et al., 2014 [40]	CTCs before and after adjuvant chemotherapy	CellSearch	CTCs detected in 21.5% of patients pre-adjuvant chemotherapy. CTCs detected in 22.1% of patients post-adjuvant chemotherapy. CTCs before and after adjuvant therapy independently associated with decreased disease-free survival and overall survival.
Pierga, J.Y., et al., 2015 [41]	CTCs in HER2+ IBC before, during, after NACT	CellSearch	CTCs at baseline independently associated with 3-year disease free survival. No CTC detected at baseline = 81% DFS. 1 or more CTC detected at baseline = 43% DFS.
Pierga, J.Y., et al., 2017 [42]	CTCs in HER2+ IBC before, during, after NACT (pooled analysis)	CellSearch	CTCs detected in 39% of patients at baseline. CTCs detected in 9% of patients after 4 cycles of chemotherapy. No correlation between CTC and pCR. CTCs detected as baseline associated with decreased 3-year DFS and OS.
Sparano, J., et al., 2018 [43]	CTCs at 5-years after diagnosis in ER+, HER2-	CellSearch	CTCs detected in 5.1% of patients 5 years after diagnosis.CTCs independently associated with increased risk of recurrence 5 years after diagnosis.
Goodman, C.R., et al., 2018 [44]	CTCs and radiotherapy in early stage breast cancer	CellSearch	At least 1 CTC and treated with radiotherapy associated with increased RFS, DFS, and OS.
Trapp, E., et al., 2019 [45]	CTCs before and 2 years after adjuvant chemotherapy	CellSearch	CTCs detected in 18.2% of patients 2 years after chemotherapy. CTCs detected 2 years after chemotherapy associated with decreased OS and DFS.
Rossi, T., et al., 2020 [46]	Copy number alterations of CTCs pre-surgery, 1-month post-surgery, and 6-months post-surgery	OncoQuick DEPArray	CTCs presented different levels of copy number alterations based on timepoint and cancer subtype. CTCs 6 months post-surgery shared copy number alterations with primary tumor.

CTCs, circulating tumor cells; DFS, disease-free survival; NACT, neoadjuvant chemotherapy; OS, overall survival; PFS, progression-free survival; RFS, recurrence-free survival.

**Table 2 ijms-23-07843-t002:** ctDNA in early breast cancer.

Authors [Reference]	Study Design	Method	Findings
Beaver, J.A., et al., 2014 [47]	PIK3CA in pre- and post-surgery plasma samples	ddPCR	ctDNA detectable in plasma. Of 15 PIK3CA mutations detected in tumors, 14 were detected in pre-surgery plasma. No mutations detected in wild-type PIK3CA plasma. Sensitivity: 93.3%, specificity: 100%
Riva, F., et al., 2017 [48]	ctDNA in TNBC before NACT, after 1 cycle, pre-surgery, post-surgery	ddPCR	ctDNA detected in 75% of patients at baseline. ctDNA decreased during NACT. Minimal decrease in ctDNA level during NACT associated with shorter DFS and OS.
Phallen, J., et al., 2017 [49]	ctDNA at primary diagnosis	TEC-Seq	ctDNA detected in 56% stage I-III BC patients. ctDNA detected had high concordance with alterations detected in tumor tissue.
Cohen, J.D., et al., 2018 [50]	ctDNA at primary diagnosis	QIASymphony CancerSEEK	33% sensitivity in breast cancer.
Garcia-Murillas, I., et al., 2019 [51]	ctDNA before, during, and after NACT	ddPCR	ctDNA detected at baseline associated with RFS. ctDNA detected during follow-up associated with increased risk of relapse.
Garcia-Murillas, I., et al., 2015 [52]	ctDNA after NACT	ddPCR	ctDNA detected at post-surgical time point or during follow-ups increased risk of relapse. Serial ctDNA predicted relapse with median lead time of 7.9 months over clinical relapse.
Rothe, F., et al., 2019 [54]	ctDNA before, at week 2, and after NACT in HER2+ patients	ddPCR	ctDNA detected in 41% of patients at baseline. ctDNA detected in 20% of patients at week 2. ctDNA detected in 5% of patients post-NACT. ctDNA detected at baseline associated with decreased pCR.
Zhang, X., et al., 2019 [55]	ctDNA before and after adjuvant chemotherapy	AmpliSeq	ctDNA detected after surgery associated with increased lymph node metasis. ctDNA positivity decreased after chemotherapy in TNBC and HER2+. ctDNA positivity persistent after chemotherapy in ER+.
Rodriguez, B.J., et al., 2019 [56]	TP53 and PIK3CA in plasma and tissue at diagnosis	SafeSEQ	Matched plasma and tumor mutations detected in 27.6% of patients at diagnosis. Four ctDNA mutations identified in plasma but not in tumor tissue. Clinicopathological features significantly associated with ctDNA detection.
Chen, Y.H., et al., 2017 [57]	ctDNA in patients with residual disease after NACT	Oncomine Research Panel	Mutations identified in tumor tissue of 33/38 patients. ctDNA detected in 4/33 patients. All 4 patients relapsed (100% specificity). 13 patients in total relapsed (31% sensitivity). ctDNA detected in patients with residual disease associated with decreased DFS.
McDonald, B.R., et al., 2019 [60]	ctDNA before, during, and after NACT	TARDIS	ctDNA detected in 100% of patients before treatment. ctDNA concentrations decreased during treatment and were lower overall for patients who achieved pCR.

ctDNA, circulating tumor DNA; ddPCR, digital drop polymerase chain reaction; NACT, neoadjuvant chemotherapy; pCR, pathologic complete response; TARDIS, targeted digital sequencing; TNBC, triple-negative breast cancer.

**Table 3 ijms-23-07843-t003:** CTCs in metastatic breast cancer.

Authors [Reference]	Study Design	Method	Findings
Martin, M., et al., 2013 [62]	CTCs at baseline and before second cycle of chemotherapy	CellSearch	*n* = 99. Detection of CTCs before the second cycle of chemotherapy is an early and powerful predictor of treatment outcome. Patients with 0–4 CTCs had a significantly better OS, PFS, and clinical benefit rate than patients with ≥5 CTCs
Bidard, F.C., et al., 2014 [63]	CTCs before, during, and after treatment	CellSearch	*n* = 911. Patients with ≥5 CTCs before treatment had a decreased PFS and OS compared to patients with <5 CTCs. An increase in CTCs 3–5 weeks and 6–8 weeks after treatment correlated with shorter PFS and OS.
Smerage, J.B., et al., 2014 [64]	CTCs before treatment and 21 days into treatment	CellSearch	*n* = 595. After 21 days of treatment, patients with increased CTCs compared to baseline were randomly assigned to receive either initial therapy or an alternate therapy. Switching cytotoxic therapies based on an increase in CTCs did not result in a longer OS compared to patients with persistently elevated CTCs.
Cristofanilli, M., et al., 2019 [65]	CTCs for stratification of patients	CellSearch	*n* = 2436. Patients who had ≥5 CTCs were classified as Stage IV aggressive, while patients with 1–4 CTCs were classified as stage IV indolent. The stage IV indolent group had a longer median OS across all disease subtypes.
Muller, V., et al., 2021 [66]	CTCs at baseline with HER2 phenotype staining	CellSearch	*n* = 1933. Detection of one of more CTCs with strong HER2 staining was associated with a shorter OS compared to patients with negative-to-moderate HER2 staining. CTC status independently predicted OS.
Bidard, F.C., et al., 2021 [67]	CTC-driven treatment vs. clinician-driven treatment	CellSearch	*n* = 78. Median PFS was slightly longer in the CTC-driven treatment arm than in the clinician-driven treatment arm.
Cabel, L., et al., 2021 [68]	CTC-based monitoring after first-line therapy	CellSearch	*n* = 207. This study failed to demonstrate the clinical utility of CTC monitoring in metastatic breast cancer due to limited accrual and compliance.

CTCs, circulating tumor cells; OS, overall survival; PFS, progression-free survival.

**Table 4 ijms-23-07843-t004:** ctDNA in metastatic breast cancer.

Authors [Reference]	Study Design	Method	Findings
Tan, G., et al., 2018 [69]	Meta-analysis of 10 different studies		*n* = 1127. There was a strong association between cfDNA and OS, DFS, and RFS. Subgroup analyses confirmed the role of cfDNA as a strong prognostic marker regardless of cfDNA analyses, sampling time, sample source, detection method, tumor stage, sample size, or area.
Shaw, J.A., et al., 2017 [70]	Comparison of cfDNA profiles to isolated CTCs	NGS and ddPCR	*n* = 5. Total cfDNA levels were significantly associated with OS. In all 5 patients, cfDNA profiles matched the mutations found in matched, isolated CTCs (apart from two additional mutations that may have been acquired with disease progression).
Fernandez-Garcia, D., et al., 2019 [71]	ctDNA with CTCs	qPCR	*n* = 94. Level of total cfDNA is a strong predictor of OS, PFS, and disease relapse (when comparing responders to non-responders). Combining CTC and cfDNA levels is a stronger biomarker for OS than the combination of CA15-3 and AP.
Murtaza, M., et al., 2015 [72]	cfDNA throughout treatment plan to evaluate clonal evolution	ddPCR and Whole Exome Sequencing	*n* = 1. Changes in serial collections of cfDNA throughout treatment correlate with different treatment responses between sites of metastatic disease, allowing cfDNA to provide real-time information of multifocal clonal evolution.
Darrigues, L., et al., 2021 [75]	cfDNA before, during, after treatment and at disease progression	ddPCR	*n* = 61. Serial analyses of cfDNA is an effective tool to measure treatment response to palbociclib and fulvestrant in ER+ metastatic breast cancer. Early variations in levels of cfDNA is a prognostic factor of PFS.
Fabrice Andre, F.S., et al., 2020 [76]	ctDNA at baseline	NGS	*n* = 1053. Genetic alterations from ctDNA can serve as potential biomarkers of response or resistance in metastatic breast cancer.
Hrebien, S., et al., 2019 [77]	ctDNA levels at baseline and early treatment	ddCPR	*n* = 59. Early changes in ctDNA dynamics were a strong indicator for PFS
Turner, N.C., et al., 2020 [78]	ctDNA before treatment	ddPCR and targeted sequencing	*n* = 1034. Concordance between ddPCR and targeted sequencing of ctDNA was 96–99%, and sensitivity of ddPCR ctDNA mutations identified in tissue sequencing was 98%. Three of the four treatment arms met or exceeded target response rate, demonstrating that ctDNA testing offers accurate and rapid genotyping that can allow the selection of mutation-directed treatments for metastatic breast cancer patients.

cfDNA, cell-free DNA; CTC, circulating tumor cell; ctDNA, circulating tumor DNA; ddPCR, digital drop polymerase chain reaction; DFS, disease-free survival; NGS, next-generation sequencing; OS, overall survival; TNBC, triple-negative breast cancer.

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
