# Peer review of "Applications of Circulating Tumor Cells and Circulating Tumor DNA in Precision Oncology for Breast Cancers"

_ijms, 2022, doi:10.3390/ijms23147843_

Round 1
Reviewer 1 Report
The authors present many data on the detection fo CTCs by the cell search sytem. This relies on EpCAM detection. Hovewer, EpCAM is down-regulated during EMT. The authors should discuss this problem as it migth lead to a lower detection number of CTCs.
Gorges TM, Tinhofer I, Drosch M, Röse L, Zollner TM, Krahn T, von Ahsen O. Circulating tumour cells escape from EpCAM-based detection due to epithelial-to-mesenchymal transition. BMC Cancer. 2012 May 16;12:178. doi: 10.1186/1471-2407-12-178. PMID: 22591372; PMCID: PMC3502112.
The above paper corroborates earlier experimental findings (EGP2 = EpCAM):
Jojović M, Adam E, Zangemeister-Wittke U, Schumacher U. Epithelial glycoprotein-2 expression is subject to regulatory processes in epithelial-mesenchymal transitions during metastases: an investigation of human cancers transplanted into severe combined immunodeficient mice. Histochem J. 1998 Oct;30(10):723-9. doi: 10.1023/a:1003486630314. PMID: 9873999.
Author Response
Thank you for bringing up this critical caveat of the CellSearch system. We briefly mentioned it earlier in the manuscript; we now included a small paragraph discussing the insensitivity of the CellSearch Immuno-magnetic system to detect changes in the epithelial marker. Please find it in the section “Drawbacks of existing liquid biopsy approaches” lines 495-511.
Reviewer 2 Report
This is a detailed review of current knowledge and use of circulating tumour biomarkers in breast cancer.
The major strengths of this review include in-depth description of previous studies that have analysed the use of circulating tumour biomarkers in prognostication with some comments on personalised medicine.
There remains a large gap in the scientific literature with regards to early diagnosis of patients with breast cancer using these techniques, which the review highlights appropriately.
The tables present a summary of key results from papers cited in this review.
Spelling/Grammar
Line 278: in the heading, 'breast' is spelt incorrectly
Line 553: 'investigating' is spelt incorrectly
Author Response
Thank you for the comments; we made the edits to the manuscript and ran it through spellcheck.